# Does patients' age predict their clinical outcomes following non-infectious epiglottitis? A systematic review

**Alaa Safia** [1,2] *, **Uday Abd Elhadi** [1,2], **Rabie Shehadeh** [1], **Raed Farhat** [1], **Majd Asakly** [1], **Nidal El Khatib** [1], **Ashraf Khater** [1], **Taiser Bishara** [1], **Saqr Massoud** [1], **Shlomo Merchavy** [1]

1 Department of Otolaryngology, Head & Neck Surgery Unit, Rebecca Ziv Medical Center, Safed, Israel,
2 True Doctor, Research Wing, Israel

* alaa.safia03@gmail.com

## Abstract

### Background

Non-infectious epiglottitis, an infrequent but significant condition, presents challenges in airway management and treatment due to its potential for rapid progression.

### Objective

To analyze differences in clinicodemographic characteristics, management strategies, and clinical outcomes between pediatric and adult cases of non-infectious epiglottitis.

### Methods

A systematic search of four databases identified 57 patient records, all diagnosed with non-infectious epiglottitis. Children (<18 years) were compared to adults (≥18 years). Differences in clinicodemographic characteristics, management strategies, and clinical outcomes were analyzed. Outcomes included intubation, complications, and intensive care unit (ICU) admission. Risk factors of these outcomes were identified through uni- and multi-variable logistic regression analyses.

### Results

Twenty-three children and 34 adults were analyzed. The presentation with stridor (56.52% vs. 14.7%), drooling (56.52% vs. 26.47%), cyanosis (17.39% vs. 0%), and sternal retraction (13.04% vs. 0%) was more common among children. Prior vaccination was evident in only 5 pediatric cases. The etiology of epiglottitis was similar across groups. Children had significantly higher chances of receiving epinephrine (34.78% vs. 8.82%), undergoing intubation (82.60% vs. 20.58%), being admitted to the ICU (56.52% vs. 17.64%), and having complications (47.82% vs. 14.70%), compared to adults. In the multivariate regression model, pediatric age was a risk factor for intubation (p = 0.015) and ICU admission (p = 0.040), while foreign body ingestion (p = 0.039) and dyspnea (p = 0.014) were predictors of intubation and complications, respectively.

**Data Availability Statement:** The data analyzed in this research are provided as supplementary file.

**Funding:** The author(s) received no specific funding for this work.

**Competing interests:** The authors have declared that no competing interests exist.

## Conclusions

The study highlights the necessity for age-specific management strategies in non-infectious epiglottitis. Understanding the distinct clinical presentations and responses in different age groups can lead to improved patient care.

## 1. Introduction

Non-infectious epiglottitis, while less common than its infectious counterpart, represents a significant clinical burden due to its potential for rapid progression and airway obstruction [1]. Managing this condition is complex and requires balancing therapeutic intervention with watchful waiting, especially considering the varied etiologies such as thermal, traumatic, or chemical injuries [1].

The role of patient age in the management of non-infectious epiglottitis is particularly critical [2]. Children, due to their anatomically smaller airways and different immune responses, may be at a higher risk for severe outcomes and thus require different management strategies compared to adults. Factors, such as the specific cause of epiglottitis and presenting symptoms significantly influence the need for airway intervention [3].

Non-infectious epiglottitis presents a significant clinical burden, particularly in terms of airway management. The necessity for airway intervention, such as intubation, is a critical concern, with varying rates of requirement based on patient age and the severity of presentation [4]. Complications associated with the disease can be severe, leading to prolonged hospital stays and increased healthcare costs. Additionally, the risk of ICU admission underscores the potential severity of this condition, highlighting the need for prompt recognition and effective management strategies to mitigate these risks.

This research addresses a gap in understanding the nuanced differences in the presentation and management of non-infectious epiglottitis across different age groups. By pooling individual patient data in a meta-analytical approach, we aim to provide an age-oriented understanding of the presentation patters, management plans, and subsequent clinical outcomes of patients with non-infectious epiglottitis.

## 2. Materials and methods

### 2.1. Research design and protocol registration

This individual participant data meta-analysis study included patients' data from available case reports, case series, and cohort studies of non-infectious epiglottitis records. The study protocol was registered on PROSPERO [protocol ID: CRD42024497541]. The study adhered to the Preferred Reporting Items for Systematic Reviews and Meta-Analyses (PRISMA) reporting guidelines for individual patient data meta-analysis (S1 File).

### 2.2. Information sources

The identification and selection of relevant patient records was done through a systematic literature search of PubMed, Scopus, Web of Science, and Google Scholar. The search included studies that were published from inception until December 22nd, 2023. Detailed search criteria for identifying relevant cases are presented in S1 Table. Our institution's librarian conducted the literature search. Moreover, a manual search step was done to ensure the inclusion of all eligible studies without missing any potentially-relevant studies. This was done by searching

similar articles on PubMed using the "similar articles" option, searching the citations of included studies, and searching Google software using the same keywords employed in the original database search.

## 2.3. Record selection (eligibility criteria)

We selected all records of patients, of all ages, diagnosed with epiglottitis of non-infectious origin (either thermal, traumatic, caustic, etc.). The records must include data on patients' clinicodemographic characteristics, management protocol (administered interventions), and clinical outcomes (mainly airway intervention–intubation and tracheostomy). When diagnosis was in doubt, patients were closely monitored in the Emergency department or ICU, administered empirical antibiotic therapy plus corticosteroid (to prevent secondary infection), and underwent further diagnostic procedures such as repeated laryngoscopy, CT, or MRI. Records were excluded if they reported infectious epiglottitis or lacked data on patients' age, etiology of epiglottitis, or clinical outcomes. Additionally, if a study did not clarify the diagnostic approach to confirm the diagnosis, it was ruled out.

## 2.4. Data extraction and methodological quality assessment

The senior author formatted the extraction sheet through Excel to fit the study's outcomes. The sheet was divided into four parts: studies' and patients' baseline data, the management plan for epiglottitis, clinical outcomes post-treatment, and the methodological quality (S2 File).

The first part focused on patients' clinicodemographic data that covered patients' age, gender, presenting symptoms, prior Hemophilus *influenzae* vaccination, blood culture (performed vs. no), and etiology of non-infectious epiglottitis. Patients were categorized, based on their age, into children (<18 years) and adults ($\geq$ 18 years). The etiology of epiglottitis was classified as traumatic, thermal (heat injury), caustic (chemical injury), foreign body ingestion, autoimmune disease-associated, smoking-related, or angioedema.

The second part focused on the management protocol, including data on administered antibiotics, steroids, epinephrine, and airway intervention, defined as either intubation or tracheostomy. Intubation was further subcategorized into immediate, delayed, and unsuccessful intubation.

The third part focused on patient outcomes, including airway intervention (primary outcome), complications, intensive care unit (ICU) admission, ICU duration (in days), length of hospital stay (LOS, in days), complications, readmission rate, and death (secondary outcomes). Complications were further classified into respiratory distress, respiratory arrest, and pulmonary edema. Additionally, the risk factors of intubation, ICU admission, and complications were investigated.

The final part focused on assessing included studies' methodological quality as per the National Institute of Health (NIH) quality assessment tool for case series. The tool was also employed in case reports [5]. Overall, a study was given a rating of good, fair, or poor quality.

## 2.5. Data analysis

Data analysis was conducted using STATA Software (Version 18). Descriptive statistics included mean and standard deviation for continuous data, and frequency and percentages for categorical data. The Shapiro-Wilk test assessed data normality. Differences between pediatric and adult cases of non-infectious epiglottitis in terms of clinicodemographic data, management plan, and clinical outcome were determined using the Chi-square test. Then, univariate and multivariable logistic regression models were designed to determine the risk

factors of airway intubation, complications, and ICU admission. All baseline data were included in the univariate models, and factors having a *P*-value below 0.25 were implemented in the multivariable models. In both models, the risk of the outcome was determined with the crude odds ratio–OR (in the univariate model) and adjusted odds ratio–aOR (in the multivariable model). Multicollinearity between covariates was investigated using the Variance Inflation Factor (VIF). Covariates with VIF scores >10 were excluded. Model fit was determined through R-squared ($R^2$). Good model fit was defined by greater $R^2$ values ($> 0.50$). Since this is an individual patient data meta-analysis, sensitivity analysis and publication bias assessment were not feasible [6]. Statistically significant differences were defined by a *P*-value of 0.05 or less. The analyzed dataset can be accessed through S3 File.

## 3. Results

### 3.1. Literature search results

The results of the literature search and study selection processes are illustrated in Fig 1. In summary, we identified 1140 records from the literature search (S4 File), of which 337 were ruled out as duplicates through EndNote Software. Following the screening of 803 titles and abstracts, only 109 articles were sought for full-text retrieval, four of which were inaccessible. The main (first and corresponding) authors of these articles were contacted, but no response was received. Sixty-five articles were then excluded because they lacked data on patients' age

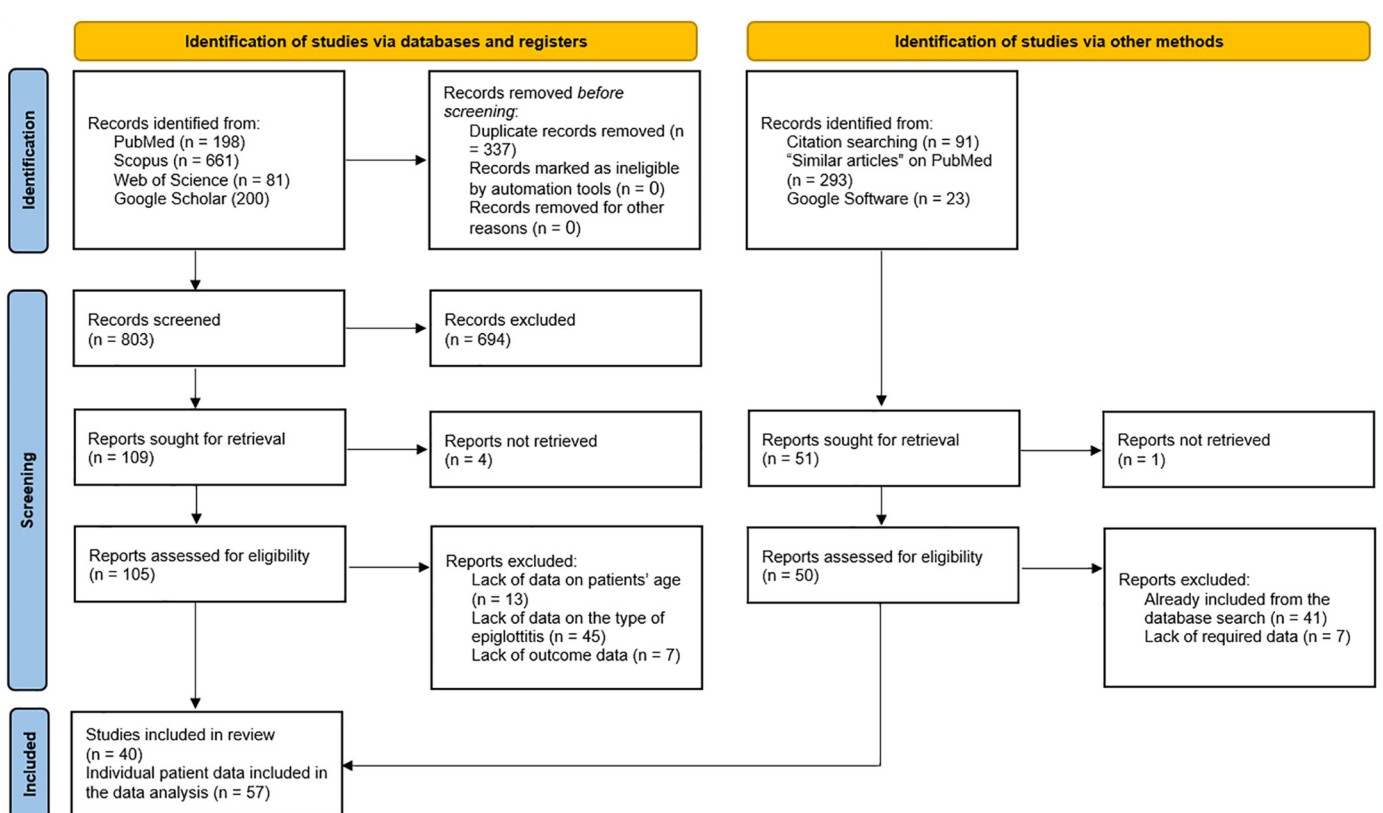

**Fig 1. A PRISMA flow diagram showing the literature search and screening results.** n: number of studies.

(13 records), type of epiglottitis (45 records), or outcomes (7 records). Finally, 40 studies were included, reporting 57 individual patient data [2, 7–45].

## 3.2. Included studies' characteristics and risk of bias

Forty studies were included, of which 7 were case series and the remaining 33 studies were individual case reports. In these studies, 57 non-infectious epiglottis cases were reported. Most patients were males (61.4%) in the adult age group (>18 years, 59.64%). A quantitative synthesis of included patients' clinicodemographic characteristics is provided in the following subsections. In terms of methodological quality, most studies had good quality and only six studies had fair quality (S2 Table).

## 3.3. Clinicodemographic characteristics of non-infectious epiglottitis

Most patients with non-infectious epiglottitis were males (35/57, 61.4%) with a mean age of 27.79 (SD = 24.59) years (Table 1). Thirty-four cases were adults and 23 were children.

Table 1. Baseline clinicodemographic characteristics of patients with non-infectious epiglottitis.

| Variable | Adults (N = 34) | Children (N = 23) | Total (N = 57) | P Value |
|---|---|---|---|---|
| Male | 18 (52.94) | 17 (73.91) | 35 (61.4) | 0.11 |
| Age–mean (SD) | 44 (18.44) | 3.82 (4.82) | 27.79 (24.59) | **0.00001** |
| Vaccinated | 0 (0) | 5 (21.73) | 5 (8.77) | **0.004** |
| Blood culture (performed) | 2 (5.88) | 9 (39.13) | 11 (19.29) | **0.002** |
| **Symptoms** | | | | |
| Stridor | 5 (14.7) | 13 (56.52) | 18 (31.57) | **0.001** |
| Drooling | 9 (26.47) | 13 (56.52) | 22 (38.59) | **0.022** |
| Dyspnea | 12 (35.29) | 10 (43.47) | 22 (38.59) | 0.533 |
| Dysphagia | 16 (47.05) | 11 (47.82) | 27 (47.36) | 0.955 |
| Irritability | 1 (2.94) | 4 (17.39) | 5 (8.77) | 0.058 |
| Hoarseness of voice | 12 (35.29) | 4 (17.39) | 16 (28) | 0.14 |
| Sore throat | 14 (41.17) | 3 (13.04) | 17 (29.82) | **0.023** |
| Odynophagia | 7 (20.58) | 1 (4.34) | 8 (14) | 0.083 |
| Angioedema | 0 (0) | 2 (8.69) | 2 (3.5) | 0.08 |
| Cyanosis | 0 (0) | 4 (17.39) | 4 (7) | **0.012** |
| Choking | 1 (2.94) | 0 (0) | 1 (1.75) | 0.407 |
| Regurgitation | 0 (0) | 1 (4.34) | 1 (1.75) | 0.22 |
| Ecchymosis | 2 (5.88) | 0 (0) | 2 (3.5) | 0.236 |
| Sternal retraction | 0 (0) | 3 (13.04) | 2 (3.5) | **0.03** |
| Coma | 0 (0) | 1 (4.34) | 1 (1.75) | 0.22 |
| **Etiology of Epiglottitis** | | | | |
| Thermal | 8 (23.52) | 8 (34.78) | 16 (28) | 0.354 |
| Caustic | 6 (17.64) | 2 (8.69) | 8 (14) | 0.34 |
| Foreign body ingestion | 10 (29.41) | 5 (21.73) | 15 (26.31) | 0.519 |
| Smoking | 4 (11.76) | 2 (8.69) | 6 (10.52) | 0.711 |
| Autoimmune disease | 2 (5.88) | 0 (0) | 2 (3.5) | 0.236 |
| Traumatic | 1 (2.94) | 2 (8.69) | 3 (5.26) | 0.34 |
| Angioedema | 2 (5.88) | 1 (4.34) | 3 (5.26) | 0.799 |
| Suicidal attempt | 1 (2.94) | 0 (0) | 1 (1.75) | 0.407 |

Values highlighted in bold indicate statistically significant differences (P < 0.05). Data are presented as numbers (percentages). N: number of cases; SD: standard deviation.

Children with non-infectious epiglottitis were significantly more likely to present with stridor (56.52% vs. 14.7%, p = 0.001), drooling (56.52% vs. 26.47%, p = 0.022), cyanosis (17.39% vs. 0%, p = 0.012), and sternal retraction (13.04% vs. 0, p = 0.03). They were also more likely to be vaccinated (21.73% vs. 0%, p = 0.004) and have blood cultures performed (39.13% vs. 5.88%, p = 0.002). On the other hand, adult patients were more likely to present with sore throat (13.04% vs. 41.17%, p = 0.023). No differences were seen between both groups regarding other presentations.

In terms of etiology, thermal injury (through hot fluids or food) was the most frequent cause of non-infectious epiglottitis (16/57, 28%), followed by foreign body ingestion (15/57, 26.31%), caustic injury (through chemicals, 8/57, 14%), and smoking (6/57, 10.52%), respectively (Table 1). The likelihood of etiological factors was similar across pediatric and adult cases of non-infectious epiglottitis (p > 0.05).

### 3.4. Management plan of non-infectious epiglottitis

During the initial presentation, most patients were administered antibiotics (36/57, 63.15%) and steroids (34/57, 59.64%), while epinephrine was given in only a minority of cases (11/57, 19.29%) (Table 2). Airway intervention was initiated in approximately half the population (26/57, 45.61%). Most cases had delayed intubation (13/57, 22.8%) compared to immediate intubation (6/57, 10.52%). Initial intubation was unsuccessful in two cases (3.5%), while tracheostomy was performed in five cases (8.77%).

Upon comparing both adult and pediatric cases, children with non-infectious epiglottitis were significantly more likely to receive epinephrine (34.78% vs. 8.82%, p = 0.015) and undergo airway intubation (82.60% vs. 20.58%, p = 0.0001) both immediate (21.73% vs. 2.94%, p = 0.023) or delayed (43.47% vs. 8.82%, p = 0.002) (Table 2).

### 3.5. Clinical outcomes of non-infectious epiglottitis

One-third (19/57) of cases were admitted to the ICU with a mean duration of 4.95 (SD = 3.25, range 1–13) days (Table 3). The mean length of hospital stay was 5.23 (SD = 3.57) days. Overall complications occurred in 16 cases (28.07%), the most frequent of which being respiratory distress (13/57, 22.8%), followed by pulmonary edema (3/57, 5.26%) and respiratory arrest (3/57, 5.26%), respectively. Only one case (1.75%) was readmitted, and three (5.26%) deaths occurred during the mean follow-up period of 19 (SD = 18.46) days.

**Table 2. The management of patients presenting with non-infectious epiglottitis.**

| Management | Adults (N = 34) | Children (N = 23) | Total (N = 57) | *P* value |
|---|---|---|---|---|
| Antibiotic | 23 (67.64) | 13 (56.52) | 36 (63.15) | 0.393 |
| Steroid | 23 (67.64) | 11 (47.82) | 34 (59.64) | 0.135 |
| Epinephrine | 3 (8.82) | 8 (34.78) | 11 (19.29) | **0.015** |
| **Airway intervention** | | | | |
| Intubation | 7 (20.58) | 19 (82.60) | 26 (45.61) | **0.0001** |
| Immediate intubation | 1 (2.94) | 5 (21.73) | 6 (10.52) | **0.023** |
| Delayed intubation | 3 (8.82) | 10 (43.47) | 13 (22.8) | **0.002** |
| Tracheostomy | 4 (11.76) | 1 (4.34) | 5 (8.77) | 0.331 |
| Unsuccessful intubation | 1 (2.94) | 1 (4.34) | 2 (3.5) | 0.777 |

Values highlighted in bold indicate statistically significant differences (P < 0.05). Data are presented as numbers (percentages). N: number of cases.

**Table 3. The clinical outcomes following the management of patients with non-infectious epiglottitis.**

| Outcomes | Adults (N = 34) | Children (N = 23) | Total (N = 57) | *P* value |
|---|---|---|---|---|
| ICU admission | 6 (17.64) | 13 (56.52) | 19 (33.33) | **0.002** |
| ICU duration (days) | 5.25 (5.31) | 4.78 (1.86) | 4.95 (3.25) | 0.875 |
| LOS (days) | 4.75 (4.00) | 5.88 (3.01) | 5.23 (3.57) | 0.466 |
| Complications | 5 (14.70) | 11 (47.82) | 16 (28.07) | **0.006** |
| Respiratory distress | 3 (8.82) | 10 (43.47) | 13 (22.8) | **0.002** |
| Pulmonary edema | 0 (0) | 3 (13.04) | 3 (5.26) | **0.03** |
| Respiratory arrest | 3 (8.82) | 0 (0) | 3 (5.26) | 0.143 |
| Readmission | 0 (0) | 1 (4.34) | 1 (1.75) | 0.22 |
| Death | 1 (2.94) | 2 (8.69) | 3 (5.26) | 0.34 |
| Follow-up (days) | 19.33 (21.34) | 18.33 (14.97) | 19 (18.46) | 0.937 |

Values highlighted in bold indicate statistically significant differences (P < 0.05). Data are presented as numbers (percentages). N: number of cases; ICU: intensive care unit; LOS: length of hospital stay.

Upon stratifying the clinical data by patients' age, children with non-infectious epiglottitis were significantly more likely to be admitted to the ICU (56.52% vs. 17.64%, p = 0.002), have complications (47.82% vs. 14.70%, p = 0.006), most commonly respiratory distress (43.47% vs. 8.82%, p = 0.002) and pulmonary edema (13.04% vs. 0%, p = 0.03) (Table 3).

### 3.6. Risk factors of intubation in non-infectious epiglottitis

In the univariate regression model, patients' age, symptoms, and clinical outcomes were significant risk factors for intubation among cases with non-infectious epiglottitis (Table 4). Children had a significantly higher risk of intubation compared to adults (OR = 18.32; 95% CI: 4.69–71.48). Other variables, such as stridor (OR = 7.87; 95% CI: 2.14–28.97), performed blood culture (OR = 7.67; 95% CI: 1.48–39.77), ICU admission (OR = 4.16; 95% CI: 1.28–13.51), complications (OR = 9.33; 95% CI: 2.26–38.50), and respiratory distress (OR = 10.63; 95% CI: 2.08–54.29) were associated with a significant increase in the risk of intubation. Other factors were insignificant predictors of ICU admission.

After accounting for the confounding effect of other covariates in the multivariable regression model, the pediatric age group–children (aOR = 9.99; 95% CI: 1.56–63.77) and foreign body ingestion (aOR = 7.41; 95% CI: 1.10–49.64) were the only significant risk factors for intubation (Table 4).

### 3.7. Risk factors of complications in non-infectious epiglottitis

In the univariate regression model, patients' age, symptoms, and management were significant risk factors for complications in non-infectious epiglottitis (Table 5). Children (OR = 5.31; 95% CI: 1.51–18.61), stridor (OR = 4.57; 95% CI: 1.33–15.70), dyspnea (OR = 6.0; 95% CI: 1.69–21.21), epinephrine administration (OR = 4.32; 95% CI: 1.08–17.14), and intubation (OR = 9.33; 95% CI: 2.26–38.50) were significant determinants of complications. Other factors were insignificant predictors of ICU admission.

After accounting for the confounding effect of other covariates in the multivariable regression model, dyspnea was the only significant risk factor for complications in patients with non-infectious epiglottitis (aOR = 9.35; 95% CI: 1.55–56.08) (Table 5).

**Table 4. Univariate and multivariable logistic regression models of risk factors of intubation in non-infectious epiglottitis.**

| Predictor | Univariate Regression Model | | | | | | Multivariable Regression Model | | | | | |
|---|---|---|---|---|---|---|---|---|---|---|---|---|
| | OR | SE | Z | P | Lower CI | Higher CI | aOR | SE | Z | P | Lower CI | Higher CI |
| **Age (Reference: Adults)** | | | | | | | | | | | | |
| Children | 18.321 | 12.727 | 4.190 | **0.0001** | 4.695 | 71.489 | 9.997 | 9.452 | 2.430 | **0.015** | 1.567 | 63.776 |
| **Gender (Reference: Female)** | | | | | | | | | | | | |
| Male | 1.853 | 1.033 | 1.110 | 0.269 | 0.621 | 5.526 | Excluded | | | | | |
| **Symptom (Reference: None)** | | | | | | | | | | | | |
| Stridor | 7.875 | 5.234 | 3.100 | **0.002** | 2.140 | 28.975 | 2.741 | 2.532 | 1.090 | 0.275 | 0.448 | 16.760 |
| Drooling | 2.444 | 1.362 | 1.600 | 0.109 | 0.820 | 7.285 | Excluded | | | | | |
| Dyspnea | 1.800 | 0.990 | 1.070 | 0.285 | 0.613 | 5.289 | Excluded | | | | | |
| Dysphagia | 0.914 | 0.487 | -0.170 | 0.866 | 0.322 | 2.598 | Excluded | | | | | |
| Hoarseness | 0.433 | 0.270 | -1.340 | 0.179 | 0.128 | 1.469 | Excluded | | | | | |
| Sore throat | 0.158 | 0.113 | -2.590 | **0.010** | 0.039 | 0.640 | Omitted (high multicollinearity) | | | | | |
| Odynophagia | 0.347 | 0.300 | -1.220 | 0.221 | 0.064 | 1.892 | Excluded | | | | | |
| Ecchymosis | 1.200 | 1.728 | 0.130 | 0.899 | 0.071 | 20.176 | Excluded | | | | | |
| **Etiology of Epiglottitis (Reference: Angioedema)** | | | | | | | | | | | | |
| Thermal | 1.815 | 1.081 | 1.000 | 0.317 | 0.565 | 5.830 | Excluded | | | | | |
| Caustic | 0.347 | 0.300 | -1.220 | 0.221 | 0.064 | 1.892 | Excluded | | | | | |
| Foreign body | 2.206 | 1.354 | 1.290 | 0.197 | 0.663 | 7.344 | 7.419 | 7.195 | 2.070 | **0.039** | 1.109 | 49.641 |
| Smoking | 0.208 | 0.235 | -1.390 | 0.165 | 0.023 | 1.908 | Excluded | | | | | |
| Trauma | 0.580 | 0.728 | -0.430 | 0.664 | 0.050 | 6.784 | Excluded | | | | | |
| **Vaccination (Reference: No)** | | | | | | | | | | | | |
| Vaccinated | 5.455 | 6.288 | 1.470 | 0.141 | 0.570 | 52.235 | 0.992 | 1.489 | -0.010 | 0.995 | 0.052 | 18.803 |
| **Blood Culture (Reference: Not performed)** | | | | | | | | | | | | |
| Culture (-ve) | 7.676 | 6.443 | 2.430 | **0.015** | 1.482 | 39.771 | 2.603 | 2.926 | 0.850 | 0.395 | 0.288 | 23.565 |
| **Management (Reference: None)** | | | | | | | | | | | | |
| Antibiotic | 1.625 | 0.909 | 0.870 | 0.386 | 0.543 | 4.865 | Excluded | | | | | |
| Steroid | 0.351 | 0.196 | -1.880 | 0.060 | 0.117 | 1.047 | Excluded | | | | | |
| Epinephrine | 4.148 | 3.075 | 1.920 | 0.055 | 0.970 | 17.738 | Excluded | | | | | |
| Tracheostomy | 5.455 | 6.288 | 1.470 | 0.141 | 0.570 | 52.235 | Excluded | | | | | |
| **Clinical outcomes (Reference: None)** | | | | | | | | | | | | |
| ICU admission | 4.167 | 2.502 | 2.380 | **0.017** | 1.284 | 13.517 | 2.901 | 2.686 | 1.150 | 0.250 | 0.473 | 17.813 |
| ICU duration (day) | 2.109 | 1.082 | 1.450 | 0.146 | 0.771 | 5.767 | Excluded | | | | | |
| LOS (day) | 1.599 | 0.401 | 1.870 | 0.061 | 0.978 | 2.615 | Excluded | | | | | |
| Complications | 9.333 | 6.749 | 3.090 | **0.002** | 2.262 | 38.508 | 5.590 | 10.285 | 0.940 | 0.350 | 0.152 | 205.890 |
| Respiratory distress | 10.633 | 8.846 | 2.840 | **0.004** | 2.082 | 54.298 | 1.727 | 3.473 | 0.270 | 0.786 | 0.034 | 88.924 |
| Respiratory arrest | 0.580 | 0.728 | -0.430 | 0.664 | 0.050 | 6.784 | Excluded | | | | | |
| Follow-up (day) | 1.008 | 0.044 | 0.180 | 0.854 | 0.925 | 1.099 | Excluded | | | | | |
| **Model Prediction Ability ($R^2 = 0.76$)** | | | | | | | | | | | | |

These variables were removed from the univariable model due to lack of enough power: irritability, angioedema, cyanosis, choking, regurgitation, coma, sternal retraction, suicide, autoimmune diseases, pulmonary edema, and readmission. Significant predictors of intubation were highlighted in bold ($P < 0.05$). OR: odds ratio; SE: standard error; P: p-value; CI: confidence interval; ICU: intensive care unit; LOS: length of hospital stay; aOR: adjusted odds ratio.

## 3.8. Risk factors of ICU admission in non-infectious epiglottitis

In the univariate regression model, patients' age, symptoms, and vaccination were significant risk factors for ICU admission in non-infectious epiglottitis (Table 6). Children (OR = 6.06; 95% CI: 1.81–20.28), irritability (OR = 9.86; 95% CI: 1.01–95.69), and

**Table 5. Univariate and multivariable logistic regression models of risk factors of complications in non-infectious epiglottitis.**

| Predictor | Univariate Model | | | | | | Multivariable Model | | | | | |
|---|---|---|---|---|---|---|---|---|---|---|---|---|
| | OR | SE | Z | *P* | Lower CI | Higher CI | aOR | SE | Z | *P* | Lower CI | Higher CI |
| **Gender (Reference: Female)** | | | | | | | | | | | | |
| Male | 2.348 | 1.544 | 1.300 | 0.194 | 0.647 | 8.519 | 4.184 | 4.060 | 1.470 | 0.140 | 0.625 | 28.024 |
| **Age (Reference: Adults)** | | | | | | | | | | | | |
| Children | 5.317 | 3.399 | 2.610 | **0.009** | 1.519 | 18.613 | 1.150 | 1.098 | 0.150 | 0.884 | 0.177 | 7.468 |
| **Symptom (Reference: None)** | | | | | | | | | | | | |
| Stridor | 4.571 | 2.878 | 2.410 | **0.016** | 1.331 | 15.701 | 2.807 | 2.783 | 1.040 | 0.298 | 0.402 | 19.595 |
| Drooling | 1.929 | 1.155 | 1.100 | 0.273 | 0.596 | 6.235 | Excluded | | | | | |
| Dyspnea | 6.000 | 3.866 | 2.780 | **0.005** | 1.697 | 21.213 | 9.350 | 8.546 | 2.450 | **0.014** | 1.559 | 56.082 |
| Dysphagia | 1.158 | 0.683 | 0.250 | 0.804 | 0.364 | 3.680 | Excluded | | | | | |
| Irritability | 1.810 | 1.746 | 0.610 | 0.539 | 0.273 | 11.992 | Excluded | | | | | |
| Hoarseness | 0.806 | 0.541 | -0.320 | 0.748 | 0.216 | 3.005 | Excluded | | | | | |
| Sore throat | 0.718 | 0.479 | -0.500 | 0.620 | 0.194 | 2.658 | Excluded | | | | | |
| Odynophagia | 0.324 | 0.360 | -1.010 | 0.311 | 0.037 | 2.869 | Excluded | | | | | |
| Angioedema | 2.667 | 3.857 | 0.680 | 0.498 | 0.157 | 45.397 | Excluded | | | | | |
| **Etiology of Epiglottitis (Reference: Angioedema)** | | | | | | | | | | | | |
| Thermal | 0.806 | 0.541 | -0.320 | 0.748 | 0.216 | 3.005 | Excluded | | | | | |
| Caustic | 0.833 | 0.730 | -0.210 | 0.835 | 0.150 | 4.636 | Excluded | | | | | |
| Foreign body | 0.909 | 0.615 | -0.140 | 0.888 | 0.241 | 3.423 | Excluded | | | | | |
| Trauma | 1.300 | 1.640 | 0.210 | 0.835 | 0.110 | 15.419 | Excluded | | | | | |
| **Vaccination (Reference: No)** | | | | | | | | | | | | |
| Vaccinated | 1.810 | 1.746 | 0.610 | 0.539 | 0.273 | 11.992 | Excluded | | | | | |
| **Blood Culture (Reference: Not performed)** | | | | | | | | | | | | |
| Culture (-ve) | 2.652 | 1.849 | 1.400 | 0.162 | 0.676 | 10.399 | 0.823 | 0.875 | -0.180 | 0.854 | 0.102 | 6.616 |
| **Management (Reference: None)** | | | | | | | | | | | | |
| Antibiotic | 0.667 | 0.401 | -0.670 | 0.501 | 0.205 | 2.169 | Excluded | | | | | |
| Steroid | 0.403 | 0.243 | -1.510 | 0.131 | 0.124 | 1.312 | 0.354 | 0.317 | -1.160 | 0.246 | 0.062 | 2.042 |
| Epinephrine | 4.320 | 3.038 | 2.080 | **0.037** | 1.089 | 17.140 | 1.959 | 2.271 | 0.580 | 0.562 | 0.202 | 19.013 |
| Intubation | 9.333 | 6.749 | 3.090 | **0.002** | 2.262 | 38.508 | 3.331 | 3.508 | 1.140 | 0.253 | 0.423 | 26.238 |
| Delayed intubation | 2.914 | 1.931 | 1.610 | 0.106 | 0.795 | 10.678 | 1.490 | 1.737 | 0.340 | 0.732 | 0.152 | 14.639 |
| Tracheostomy | 1.810 | 1.746 | 0.610 | 0.539 | 0.273 | 11.992 | Excluded | | | | | |
| **Clinical outcomes (Reference: None)** | | | | | | | | | | | | |
| ICU admission | 1.292 | 0.796 | 0.420 | 0.677 | 0.386 | 4.321 | Excluded | | | | | |
| ICU duration (day) | 1.160 | 0.247 | 0.700 | 0.484 | 0.765 | 1.760 | Excluded | | | | | |
| LOS (day) | 1.101 | 0.153 | 0.700 | 0.487 | 0.839 | 1.445 | Excluded | | | | | |
| Follow-up (day) | 0.777 | 0.216 | -0.910 | 0.364 | 0.451 | 1.340 | Excluded | | | | | |
| **Model Prediction Ability ($R^2$ = 0.56)** | | | | | | | | | | | | |

These variables were removed from the univariate model because of lack of enough power: cyanosis, smoking, autoimmune diseases, suicide, immediate intubation, unsuccessful intubation, readmission, and death. Significant predictors of intubation were highlighted in bold ($P < 0.05$). OR: odds ratio; SE: standard error; P: p-value; CI: confidence interval; ICU: intensive care unit; LOS: length of hospital stay; aOR: adjusted odds ratio.

vaccination (OR = 9.86; 95% CI: 1.01–95.69). Other factors were insignificant predictors of ICU admission.

After controlling for potential confounders in the multivariable regression model, the pediatric age (children) was the only significant risk factor for ICU admission in patients with non-infectious epiglottitis (aOR = 4.97; 95% CI: 1.07–22.95) (Table 6).

**Table 6. Univariate and multivariable logistic regression models of risk factors of ICU admission in non-infectious epiglottitis.**

| Predictor | Univariate Model | | | | | | Multivariable Model | | | | | |
|---|---|---|---|---|---|---|---|---|---|---|---|---|
| | OR | SE | Z | P | Lower CI | Higher CI | aOR | SE | Z | P | Lower CI | Higher CI |
| **Age (Reference: Adults)** | | | | | | | | | | | | |
| Children | 6.067 | 3.736 | 2.930 | **0.003** | 1.814 | 20.285 | 4.976 | 3.882 | 2.060 | **0.040** | 1.079 | 22.958 |
| **Gender (Reference: Female)** | | | | | | | | | | | | |
| Male | 1.118 | 0.648 | 0.190 | 0.847 | 0.359 | 3.484 | Excluded | | | | | |
| **Symptom (Reference: None)** | | | | | | | | | | | | |
| Stridor | 2.900 | 1.732 | 1.780 | 0.075 | 0.900 | 9.349 | 1.495 | 1.094 | 0.550 | 0.583 | 0.356 | 6.274 |
| Drooling | 1.731 | 0.991 | 0.960 | 0.338 | 0.563 | 5.318 | Excluded | | | | | |
| Dyspnea | 1.247 | 0.715 | 0.380 | 0.701 | 0.405 | 3.837 | Excluded | | | | | |
| Dysphagia | 1.000 | 0.563 | 0.000 | 1.000 | 0.332 | 3.013 | Excluded | | | | | |
| Irritability | 9.867 | 11.437 | 1.970 | **0.048** | 1.017 | 95.690 | Omitted (high multicollinearity) | | | | | |
| Hoarseness | 0.578 | 0.383 | -0.830 | 0.407 | 0.158 | 2.115 | Excluded | | | | | |
| Sore throat | 0.513 | 0.338 | -1.010 | 0.310 | 0.141 | 1.864 | Excluded | | | | | |
| Odynophagia | 0.246 | 0.273 | -1.260 | 0.206 | 0.028 | 2.164 | Omitted (high multicollinearity) | | | | | |
| Cyanosis | 0.648 | 0.772 | -0.360 | 0.716 | 0.063 | 6.685 | Excluded | | | | | |
| Sternal Retraction | 1.000 | 1.258 | 0.000 | 1.000 | 0.085 | 11.778 | Excluded | | | | | |
| **Etiology of Epiglottitis (Reference: Angioedema)** | | | | | | | | | | | | |
| Thermal | 1.880 | 1.146 | 1.030 | 0.301 | 0.569 | 6.210 | Excluded | | | | | |
| Caustic | 0.627 | 0.546 | -0.540 | 0.592 | 0.114 | 3.452 | Excluded | | | | | |
| Foreign body | 1.000 | 0.638 | 0.000 | 1.000 | 0.286 | 3.492 | Excluded | | | | | |
| Smoking | 2.188 | 1.904 | 0.900 | 0.369 | 0.397 | 12.048 | Excluded | | | | | |
| Trauma | 1.000 | 1.258 | 0.000 | 1.000 | 0.085 | 11.778 | Excluded | | | | | |
| **Vaccination (Reference: No)** | | | | | | | | | | | | |
| Vaccinated | 9.867 | 11.437 | 1.970 | **0.048** | 1.017 | 95.690 | 3.496 | 4.471 | 0.980 | 0.328 | 0.285 | 42.876 |
| **Blood Culture (Reference: Not performed)** | | | | | | | | | | | | |
| Culture (-ve) | 1.905 | 1.305 | 0.940 | 0.347 | 0.497 | 7.294 | Excluded | | | | | |
| **Management (Reference: None)** | | | | | | | | | | | | |
| Antibiotics | 2.036 | 1.254 | 1.150 | 0.248 | 0.609 | 6.810 | 2.683 | 1.935 | 1.370 | 0.171 | 0.653 | 11.030 |
| Steroids | 1.247 | 0.721 | 0.380 | 0.703 | 0.402 | 3.871 | Excluded | | | | | |
| Epinephrine | 1.905 | 1.305 | 0.940 | 0.347 | 0.497 | 7.294 | Excluded | | | | | |
| Tracheostomy | 0.472 | 0.546 | -0.650 | 0.516 | 0.049 | 4.546 | Excluded | | | | | |
| **Clinical outcomes (Reference: None)** | | | | | | | | | | | | |
| LOS (days) | 1.201 | 0.178 | 1.230 | 0.218 | 0.898 | 1.607 | Omitted (high multicollinearity) | | | | | |
| Complications | 1.292 | 0.796 | 0.420 | 0.677 | 0.386 | 4.321 | Excluded | | | | | |
| Respiratory distress | 1.339 | 0.878 | 0.450 | 0.656 | 0.371 | 4.840 | Excluded | | | | | |
| Respiratory arrest | 1.000 | 1.258 | 0.000 | 1.000 | 0.085 | 11.778 | Excluded | | | | | |
| Follow-up (days) | 0.950 | 0.060 | -0.810 | 0.417 | 0.838 | 1.076 | Excluded | | | | | |
| **Model Prediction Ability (R2 = 0.79)** | | | | | | | | | | | | |

These variables were removed from the univariate model due to lack of enough power: choking, regurgitation, ecchymosis, coma, readmission, pulmonary edema, suicide, autoimmune diseases, and angioedema. Significant predictors of intubation were highlighted in bold ($P < 0.05$). OR: odds ratio; SE: standard error; P: p-value; CI: confidence interval; ICU: intensive care unit; LOS: length of hospital stay; aOR: adjusted odds ratio.

## 4. Discussion

This individual patient data meta-analysis of cases of non-infectious epiglottitis provides insightful comparisons between pediatric and adult presentations, management strategies, and clinical outcomes. The notable age gap between the adult and pediatric groups is crucial. This

age-related variation might be indicative of different susceptibility or exposure to causative factors in non-infectious epiglottitis [46, 47].

In our study, we noted that pediatric cases, compared to adults, had a greater chance of presenting with stridor (by 1.6 folds), drooling (by 0.44 folds), cyanosis (by 4 folds). The higher incidence of stridor and drooling in children (56.52% and 56.52%) compared to adults (14.7% and 26.47%) suggests more pronounced upper airway involvement in pediatric cases. This is further evidenced by the higher rate of blood culture performance in children (39.13% vs. 5.88% in adults, 350% difference), indicating a more aggressive diagnostic approach in this group. The higher prevalence of specific symptoms like stridor and drooling in children could be reflective of their anatomical and physiological differences. This aspect is vital for clinicians in terms of early recognition and differential diagnosis.

Assessing the impact of vaccination status on the incidence of non-infectious epiglottitis, especially in the pediatric population, can provide insights into preventive strategies [48]. Moreover, the specific etiologies (thermal, traumatic, caustic, smoking-induced) and their frequency in each age group offer a deeper understanding of risk factors and potential protective measures. In our study, we noted that pediatric cases were more likely to be vaccinated with Hemophilus influenza vaccine (350% difference) and undergo blood culture analysis (500% difference).

The disparity in blood culture performance between the two groups might reflect differing clinical approaches or variations in symptom severity. Understanding these differences is key to optimizing diagnostic protocols. Pediatric patients had a higher rate of blood culture performance (39.13% vs. 5.88% in adults), highlighting the need to rule out secondary bacterial infections more aggressively in this age group. Implementing routine blood cultures in pediatric patients with severe symptoms can facilitate early identification and appropriate treatment of concurrent infections, thereby improving patient outcomes. For adults, a more selective approach based on clinical presentation and risk factors may prevent unnecessary antibiotic use and associated resistance. These variations suggest the necessity for age-specific clinical guidelines that balance thorough evaluation in high-risk pediatric patients with a judicious approach in adults, ultimately enhancing patient care and resource utilization.

The use of antibiotics and steroids, in our study, was prominent in both groups, reflecting a tendency to treat with broad-spectrum therapies despite the non-infectious etiology [49]. Interestingly, a trend towards an increase use of epinephrine in pediatric cases was observed (166% difference compared to adults). This could be attributed to the higher incidence of acute symptoms like stridor in the pediatric population. Additionally, the use of steroids, known for their anti-inflammatory properties, suggests an effort to manage airway swelling, a critical concern in epiglottitis. The higher use of epinephrine in children, in our study, aligns with the more acute presentations observed in this group. Epinephrine, commonly used for its rapid vasoconstrictive and bronchodilatory effects [50], could be crucial in managing acute airway obstruction symptoms, which are more pronounced in pediatric cases. These findings can pave the way for symptom-oriented approach in treating non-infectious epiglottitis, with a keen emphasis on addressing airway management challenges, particularly in pediatric patients.

In our study, the stark contrast in intubation rates (82.60% in children vs. 20.58% in adults; 166% difference) and ICU admissions (56.52% in children vs. 17.64% in adults; 116% difference) highlights the more severe clinical course in pediatric cases. However, the ICU stay duration did not significantly differ between the groups, suggesting that once stabilized, the recovery trajectory is similar across ages. The significantly higher intubation rates in children point to a more acute and severe presentation in the pediatric population. This could be due to

the smaller airway size in children, making them more susceptible to obstruction. Furthermore, pediatric cases are significantly associated with 120% difference in the probability of complications, most commonly respiratory distress (233% difference) and pulmonary edema (300% difference), compared to adults.

Based on our meta-regression model, pediatric age group was a significant determinant of both airway intubation and ICU admission, even after accounting for all other confounders. In particular, children had a greater risk of intubation and ICU admission by 9.99 and 4.97 folds, respectively. That being said, patients' age was not a predictor of complications. These findings highlight the necessity of a timely and proper management of children with presentations suggestive of non-infectious epiglottitis. Additionally, foreign body ingestion was associated with a significant increase in the risk of intubation by 7.19 folds. Meanwhile, dyspnea was the only significant determinant of complications, accounting for a 9.35-fold increase. The findings of these models highlight the most important factors that need be considered upon initiating a treatment plan, while recognizing that high-risk groups should receive an appropriate level of support early in their treatment course with more intensive monitoring and potentially-different therapeutic strategies.

Our findings indicate that non-infectious epiglottitis manifests more severely in children, necessitating more aggressive management. The higher rates of intubation and epinephrine use in pediatric cases compared to adults may indicate either a more aggressive disease course or a more proactive management approach. The increased rate of both interventions (intubation: 82.60% vs. 20.58%, epinephrine: 34.78% vs. 8.82%) in the pediatric age group may be indicative of the inherently more severe presentation in children, who are more susceptible to rapid airway obstruction due to their anatomically smaller airways and different immune responses. However, it is also possible that clinicians adopt a more aggressive management approach in pediatric patients to preempt potential complications. The higher incidence of acute symptoms such as stridor and drooling in children may prompt earlier and more intensive interventions. The frequent use of broad-spectrum therapies highlights a potential need for refining guidelines to emphasize etiology-specific treatments.

## 4.1. Clinical relevance

This study highlights significant age-related differences in the presentation and management of non-infectious epiglottitis. Pediatric patients demonstrate more severe clinical presentations, necessitating more frequent interventions such as intubation and epinephrine administration. These findings underscore the critical need for age-specific diagnostic and management protocols to improve patient outcomes. For pediatric patients, more aggressive monitoring and timely intervention are essential to prevent rapid airway obstruction and associated complications. In contrast, a more selective approach in adults may reduce unnecessary interventions and healthcare costs. Recognizing these differences and tailoring treatment strategies accordingly can lead to better resource utilization, more effective management, and ultimately improved patient care across different age groups.

## 4.2. Limitations

In addition to providing the greatest body of evidence in non-infectious epiglottitis, our study highlights the age-related differences in presentation and clinical management and the risk of negative outcome. However, our study has several potential limitations, the most important of which is the small sample size, which made it difficult to examine the impact of prior vaccination on patient outcomes, which is expected to lower the probability of presenting with acute epiglottitis drastically. Additionally, the inclusion of six fair-quality studies, while necessary to

maximize the dataset in this rare condition, may introduce some variability. However, given the predominance of good-quality studies, we believe the findings remain robust and reflective of clinical realities.

## 5. Conclusions

This individual patient data meta-analysis underscores the need for a more targeted approach in dealing with non-infectious epiglottitis based on patients' age. Pediatric patients are more likely to present with severe symptoms, have a higher probability of negative clinical outcomes, and increase the risk of intubation and intensive care unit admission.

## Supporting information

**S1 Table. The detailed search criteria employed in the literature search.**
(DOCX)

**S2 Table. The methodological quality of included studies based on the NIH quality assessment tool for case series/case reports.**
(DOCX)

**S1 File. The filled-out PRISMA checklist.**
(PDF)

**S2 File. The designed data extraction sheet for this research.**
(XLSX)

**S3 File. The analyzed dataset in this research.**
(DTA)

**S4 File. A complete list of articles retrieved through the literature search along with the reasons for exclusion.**
(XLSX)

## Author Contributions

**Conceptualization:** Alaa Safia, Nidal El Khatib, Shlomo Merchavy.

**Data curation:** Uday Abd Elhadi, Rabie Shehadeh, Raed Farhat, Majd Asakly, Nidal El Khatib, Taiser Bishara, Saqr Massoud.

**Formal analysis:** Alaa Safia.

**Investigation:** Raed Farhat, Majd Asakly, Nidal El Khatib, Ashraf Khater, Taiser Bishara, Saqr Massoud.

**Methodology:** Alaa Safia, Uday Abd Elhadi, Rabie Shehadeh, Raed Farhat, Majd Asakly, Nidal El Khatib, Ashraf Khater, Taiser Bishara, Saqr Massoud, Shlomo Merchavy.

**Project administration:** Shlomo Merchavy.

**Resources:** Shlomo Merchavy.

**Software:** Alaa Safia, Majd Asakly, Saqr Massoud, Shlomo Merchavy.

**Supervision:** Alaa Safia, Shlomo Merchavy.

**Validation:** Majd Asakly, Ashraf Khater, Saqr Massoud, Shlomo Merchavy.

**Visualization:** Majd Asakly, Ashraf Khater, Saqr Massoud.

**Writing – original draft:** Uday Abd Elhadi, Rabie Shehadeh, Raed Farhat, Nidal El Khatib.

**Writing – review & editing:** Alaa Safia, Taiser Bishara, Shlomo Merchavy.

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
