## [Decision Letter · Decision Letter 0]

23 Jul 2024

PONE-D-24-03809Does Patients’ Age Predict Their Clinical Outcomes Following Non-Infectious Epiglottitis? A Systematic ReviewPLOS ONE

Dear Dr. Safia,

Thank you for submitting your manuscript to PLOS ONE. After careful consideration, we feel that it has merit but does not fully meet PLOS ONE’s publication criteria as it currently stands. Therefore, we invite you to submit a revised version of the manuscript that addresses the points raised during the review process.

We look forward to receiving your revised manuscript.

Kind regards,

Sethu Thakachy Subha, M.S

Academic Editor

PLOS ONE

2. In this instance it seems there may be acceptable restrictions in place that prevent the public sharing of your minimal data. However, in line with our goal of ensuring long-term data availability to all interested researchers, PLOS’ Data Policy states that authors cannot be the sole named individuals responsible for ensuring data access (http://journals.plos.org/plosone/s/data-availability#loc-acceptable-data-sharing-methods).

Reviewers' comments:

Reviewer's Responses to Questions

**Comments to the Author**

1. Is the manuscript technically sound, and do the data support the conclusions?

Reviewer #1: Yes

2. Has the statistical analysis been performed appropriately and rigorously? 

Reviewer #1: Yes

3. Have the authors made all data underlying the findings in their manuscript fully available?

Reviewer #1: Yes

4. Is the manuscript presented in an intelligible fashion and written in standard English?

Reviewer #1: Yes

5. Review Comments to the Author

Reviewer #1: Thanks for this work.

Section 2.3- What was the decision where diagnosis was in doubt?

3.5- please give range of ICU stay length

4- 160% is a difference rather than an increase, as is 350%; please modify.

How could the differences in blood cultures influence the optimisation of diagnostic protocols

How do we know that the increased intubation and epinephrine represents more aggreeive disease rather than more aggressive management?

This is an interesting report, however the associations drawn need to conclude with a summary of clinical relevance.

6. PLOS authors have the option to publish the peer review history of their article (what does this mean?). If published, this will include your full peer review and any attached files.

Reviewer #1: No

---

## [Author Response · Author response to Decision Letter 0]

5 Aug 2024

Dear Editor and Reviewers,

Thank you so much for your constructive feedback and all of the great submission.

We have carefully responded to each of the raised comments in a 3-columned Table format. The first column covered the raised comments, the second covered our responses, and the third covered the applied edits to our manuscript.

The file was uploaded as "Response Table".

If you have any further questions, we would be happy to address them.

Thank you again for your time and effort in reviewing our manuscript.

---

## [Decision Letter · Decision Letter 1]

6 Jan 2025

PONE-D-24-03809R1Does Patients’ Age Predict Their Clinical Outcomes Following Non-Infectious Epiglottitis? A Systematic ReviewPLOS ONE

Dear Dr. Safia,

Thank you for submitting your manuscript to PLOS ONE. After careful consideration, we feel that it has merit but does not fully meet PLOS ONE’s publication criteria as it currently stands. Therefore, we invite you to submit a revised version of the manuscript that addresses the points raised during the review process.

We look forward to receiving your revised manuscript.

Kind regards,

John Minh Le, MD, DDS

Academic Editor

PLOS ONE

Journal Requirements:

Reviewers' comments:

Reviewer's Responses to Questions

**Comments to the Author**

1. If the authors have adequately addressed your comments raised in a previous round of review and you feel that this manuscript is now acceptable for publication, you may indicate that here to bypass the “Comments to the Author” section, enter your conflict of interest statement in the “Confidential to Editor” section, and submit your "Accept" recommendation.

Reviewer #2: (No Response)

2. Is the manuscript technically sound, and do the data support the conclusions?

Reviewer #2: Yes

3. Has the statistical analysis been performed appropriately and rigorously? 

Reviewer #2: Yes

4. Have the authors made all data underlying the findings in their manuscript fully available?

Reviewer #2: Yes

5. Is the manuscript presented in an intelligible fashion and written in standard English?

Reviewer #2: Yes

6. Review Comments to the Author

Reviewer #2: The study was done through a systematic literature search which included studies from inception until December 22nd, 2023. The study aimed at analyzing differences in clinicodemographic characteristics, management strategies and clinical outcomes between pediatric and adult cases of non-infectious epiglottitis. The study involved the search of 4 databases, which yielded 40 studies and reported 57 individual patient data.

Overall, it is a well-thought out and executed research manuscript capable of contributing to the body of knowledge regarding the management of non-infectious epiglottitis. However, I have the following observations,

1. Methods: The study utilized mostly studies that were rated as being good quality. However, six of the studies utilized in the study were deemed fair quality. A factor that might have affected the overall quality of the manuscript.

2. References: The referencing style appears generally fine. However, it appears the conference proceedings cited (references 22) was not properly cited. The approach and checklist suggested by the University of Queensland library (guides.library.uq.edu.au), might be helpful.

3. The phrasing of the language used in the entire document could benefit from a more professional grammatical input.

4. I found the title, study design, statistics, figures and tables quite satisfactory.

7. PLOS authors have the option to publish the peer review history of their article (what does this mean?). If published, this will include your full peer review and any attached files.

Reviewer #2: **Yes: **Stephen Adebola

---

## [Author Response · Author response to Decision Letter 1]

15 Jan 2025

Dear Editor and Reviewer,

Thank you for your effort in handling and reviewing my submission.

The comments were very helpful and definitely improved the quality of the manuscript.

I have uploaded a word document highlighting each comment raised, our response to it, and all applied changes to the manuscript.

IF you have any questions or further requests, we would be glad to address them.

Best,

Alaa Safia

---

## [Editor Report · Decision Letter 2]

21 Jan 2025

Does Patients’ Age Predict Their Clinical Outcomes Following Non-Infectious Epiglottitis? A Systematic Review

PONE-D-24-03809R2

Dear Dr. Safia,

We’re pleased to inform you that your manuscript has been judged scientifically suitable for publication and will be formally accepted for publication once it meets all outstanding technical requirements.

Kind regards,

John Minh Le, MD, DDS

Academic Editor

PLOS ONE
---

## [Editor Report · Acceptance letter]

30 Jan 2025

PONE-D-24-03809R2 

PLOS ONE

Dear Dr. Safia, 

I'm pleased to inform you that your manuscript has been deemed suitable for publication in PLOS ONE. Congratulations! Your manuscript is now being handed over to our production team.

Kind regards, 

on behalf of

Dr. John Minh Le 

Academic Editor

PLOS ONE